# Uptake of Flaxseed Dietary Linusorbs Modulates Regulatory Genes Including Induction of Heat Shock Proteins and Apoptosis

**DOI:** 10.3390/foods11233761

**Published:** 2022-11-22

**Authors:** Youn Young Shim, Timothy J. Tse, AkalRachna K. Saini, Young Jun Kim, Martin J. T. Reaney

**Affiliations:** 1Department of Food and Bioproduct Sciences, University of Saskatchewan, Saskatoon, SK S7N 5A8, Canada; 2Prairie Tide Diversified Inc., Saskatoon, SK S7J 0R1, Canada; 3Department of Integrative Biotechnology, Biomedical Institute for Convergence at SKKU (BICS), Sungkyunkwan University, Suwon 16419, Republic of Korea; 4Department of Food and Biotechnology, Korea University, Sejong 30019, Republic of Korea; 5Guangdong Saskatchewan Oilseed Joint Laboratory, Department of Food Science and Engineering, Jinan University, Guangzhou 510632, China

**Keywords:** flaxseed, linusorb, cyclic peptide, heat shock protein, apoptosis, anti-cancer

## Abstract

Flaxseed (*Linum usitatissimum* L.) is gaining popularity as a superfood due to its health-promoting properties. Mature flax grain includes an array of biologically active cyclic peptides or linusorbs (LOs, also known as cyclolinopeptides) that are synthesized from three or more ribosome-derived precursors. Two flaxseed orbitides, [1–9-NαC]-linusorb B3 and [1–9-NαC]-linusorb B2, suppress immunity, induce apoptosis in a cell line derived from human epithelial cancer cells (Calu-3), and inhibit T-cell proliferation, but the mechanism of LO action is unknown. LO-induced changes in gene expression in both nematode cultures and human cancer cell lines indicate that LOs promoted apoptosis. Specific evidence of LO bioactivity included: (1) distribution of LOs throughout the organism after flaxseed consumption; (2) induction of heat shock protein (HSP) 70A, an indicator of stress; (3) induction of apoptosis in Calu-3 cells; and (4) modulation of regulatory genes (determined by microarray analysis). In specific cancer cells, LOs induced apoptosis as well as HSPs in nematodes. The uptake of LOs from dietary sources indicates that these compounds might be suitable as delivery platforms for a variety of biologically active molecules for cancer therapy.

## 1. Introduction

Linusorbs (LOs) isolated from flaxseed oil are naturally occurring cyclo-oligopeptides (9–11 amino acid residues) with low molecular weights (~1 kDa) [1]. Since 1959 [2], 11 natural LOs have been identified in flaxseed oil, roots, and seed [3,4,5,6], and another 28 have been synthetically produced [7]. These LOs have been investigated for their range of therapeutic bioactivity (e.g., antioxidant, anti-inflammatory, and anti-cancer effects) [8,9,10] due to their high stability, potency, and bioavailability [11,12,13]. For instance, [1–9-NαC]-linusorb B3 (LOB3) inhibits cholate uptake in hepatocytes, thus potentially mitigating liver damage [14]. LOB3 also suppresses phosphatase activity and thereby inhibits the activation and proliferation of T-lymphocytes [15,16]. T-lymphocyte inhibition, induced by LOs, has exhibited immunosuppressant effects (e.g., delayed hypersensitivity response, postponed skin allograft rejection, suppressed post-adjuvant arthritis, and suppression of hemolytic anemia) [17,18] and cytoprotective effects [19,20].

The immunomodulatory activity of LOB3 has previously been investigated due to similarities in its nonapeptide sequence to bovine colostrum sequences [15,21]. For example, in vitro and in vivo in tests involving human lymphocyte cell lines and Jerne plaque-forming cell assays revealed immunosuppressive activity of LOB3 in both humoral and cellular immune responses [15,16]. The bioactive effects of LOB3 in in vitro studies showed the ability of this compound in delaying skin allograft rejection, inhibiting human lymphocyte proliferation, and alleviating post-adjuvant polyarthritis and hemolytic anemia in mice [15]. Low concentrations of LOB3 were also capable of inducing T and B cell proliferation, acquisition of activation antigens, and immunoglobulin (Ig) synthesis [16], similar to the biological effect of the immunosuppressant drug cyclosporine (CsA).

LOB3 has previously demonstrated non-toxic effects in both rats and mice when administered orally and intravenously [18]. The combined characteristics of the strong immunosuppressive activity and low toxicity make LOB3 an immunosuppressive drug candidate. Other LOs and their analogs have also been investigated for their immunosuppressant activity. For example, [1–9-NαC]-linusorb B2 (LOB2, also known as CLB) has also been observed to induce the proliferation of human peripheral blood lymphocytes such as CsA [3]. Similarly, [1–8-NαC], [1-(*R*_s_, *S*_s_)-MetO]-linusorb B1 (LO1OB1, also known as CLE) could also induce lymphocyte proliferation in mice [3]. Interestingly, chemical analogs of LOB3 exhibited immunosuppressant effects as well [18,22,23]. However, LOB3 demonstrated the highest activity [18,22,23]. Nonetheless, the biological activity of these analogs was influenced by their peptide sequence, with the presence of –Pro-Xxx-Phe– sequence, where Xxx denotes a hydrophobic, aliphatic, or aromatic residue contributing to a greater biological response [22,23,24] (Figure 1).

Heat shock proteins (HSP) are often expressed when stressors denature an organism’s protein. In order to stabilize denaturing proteins, chaperones (e.g., HSPs) are expressed to assist in refolding denatured proteins while facilitating the recycling of irreversibly denatured proteins [25].

The rapid expression of HSPs by *Caenorhabditis elegans* has been used as a model system to study stress responses [26]. These proteins maintain cell function and survival while playing critical roles in an organism’s ability to adapt to biotic and abiotic stressors and facilitate recovery after the removal of stressors [25,27]. For example, 70 kDa HSP (HSP 70) are highly conserved and ubiquitous chaperones that are involved in preventing protein aggregation and refolding of denatured protein [25] during cellular stress. In part, they regulate heat shock responses through mitogen-activated protein kinase (MAPK) signaling [28]. These HSP proteins have also been well-characterized in *C. elegans* [26].

The overall focus of this study was to research the unknown biological activities of LOs, including: (i) its effects on HSP 70A expression on *C. elegans* as an indicator of stress; (ii) tissue distribution following flaxseed consumption; (iii) induction of apoptosis in cancer cell lines; and (iv) its effect on apoptotic gene expression.

## 2. Materials and Methods

### 2.1. Materials

Whole flaxseed, flaxseed oil, and purified LOs (Table 1; Figure 1) from *Linum usitatissimum* L. (var. CDC Bethune) were generous gifts from Prairie Tide Diversified Inc. (Saskatoon, SK, Canada). Whole flaxseed was ground using a coffee grinder for 1 min (Model 80335, Hamilton Beach Brands, Inc., Glen Allen, VA, USA), and the flaxseed meal was then used in the feeding experiment, further described below.

### 2.2. Conformation of LOs in an Aqueous-DMSO Solution

Circular dichroism (CD) spectroscopy was performed on LO samples using a fresh solution of aqueous-dimethylsulfoxide solution (H_2_O/DMSO, 3%, *v*/*v*). CD measurements were completed using a Pistar 180 spectrometer (Applied Photophysics Ltd., Leatherhead, UK). Sample measurements were conducted at room temperature using a 0.01 cm cuvette (Hellma 106-QSP), and samples were scanned between 200–260 nm at 0.5 nm increments, with a scan rate of 10 nm/min. The instrument was calibrated using (+)-10-camphorsulfonic acid (290.5 nm wavelength). Experiments were performed in triplicate and are reported as mean ± standard deviation (SD). The secondary structure content of the samples was evaluated with the CDNN CD Spectroscopy Deconvolution software version 2.0.3.188 [30]. Molar ellipticities were calculated from molecular masses of 977.26–1074.37 Da for LOs. All spectra were corrected by background subtraction of H_2_O/DMSO (3%, *v*/*v*).

### 2.3. LOs in Pig and Fish Adipose

Swine and rainbow trout (*Oncorhynchus mykiss*) samples were obtained from previous dietary studies conducted by Juárez et al. (2010) [31] and Drew et al. (2007) [32], respectively. In trout diets, fish were fed a blend of 35% flaxseed oil with canola oil. The oil blend was included in the diet at 20% of the ration. Swine were fed a diet consisting of 15% flaxseed meal that was combined with other ingredients then extruded. Swine fat samples were taken from an 8 mm biopsy punch at the 10th rib, approximately 5 cm from the backbone [31]. The tissue samples were then extracted with chloroform and the extracts were dried, taken up in acetonitrile, and analyzed via direct loop injections using an Agilent 1100-series high-performance liquid chromatography (HPLC, Agilent, Santa Clara, CA, USA) equipped with a time-of-flight mass spectrometer (ToF-MS). Each sample extract was injected at a flow rate of 50 μL/min. The mobile phase consisted of 95% water: methanol containing 0.1% formic acid, and 5% acetonitrile.

### 2.4. Induction of HSP 70A in Nematodes

Gene expression of HSP 70A in nematodes was completed as described in Saini et al. (2011) [33]. Briefly, *C. elegans* N2 strains were cultured in the dark and at room temperature on sterile plates (10 mm; VWR, Edmonton, AB, Canada) containing 10% bacteriological agar (Sigma-Aldrich, Oakville, ON, Canada) and autoclaved Baker’s yeast (1%, *w*/*v). C. elegans* were sub-cultured every 14 days to fresh plates. Exposure experiments began using *C. elegans*, which were two weeks old. These cultures were exposed for 2 h to either DMSO (negative control) or various concentrations (0.1, 1.0, 10.0, and 100.0 µM) of LOB3 dissolved in DMSO. After exposure, these organisms were then re-cultured to assess the effects of LOB3 on HSP 70A gene expression. An unamended culture medium was used as the control treatment. After treatment, cultures were centrifuged at 400× *g* for 10 min at 4 °C. The pellet was retained and stored at −80 °C until further analysis.

### 2.5. Induction of Apoptosis in Human Lung Cancer Cell Line

#### 2.5.1. Cell Culture

Human bronchial epithelial adenocarcinoma cell line (Calu-3) was obtained from the American Type Culture Collection (ATCC, Manassas, VA, USA). Calu-3 cell line was cultured in plastic culture flasks containing Modified Eagle’s Medium (MEM) and supplemented with 10% fetal bovine serum, glutamine (4 mM), and penicillin (100 U/mL)-streptomycin (100 μg/mL) solution. Cells were cultured to a density of 10^6^ cells/mL before adding fresh media (on day 3). The cells were then grown at 37 °C under 5% CO_2_, and 100% humidity.

#### 2.5.2. Chemical Treatments

Effects of LOs were tested on Calu-3 cells at different concentrations (0–100 μM). Camptothecin (CPT) (4 μM; Sigma-Aldrich, Oakville, ON, Canada) and DMSO were used as positive and negative controls, respectively. CPT is a cytotoxic quinoline alkaloid that has demonstrated anti-cancer effects due to the inhibition of topoisomerase I, an enzyme involved in DNA replication and repair.

#### 2.5.3. Ribonucleic Acid (RNA) Isolation and Quantitative Real-Time Reverse-Transcriptase Polymerase Chain Reaction (qRT-PCR)

RNeasy^®^ Mini kits (Qiagen, Mississauga, ON, Canada) were used to extract total RNA from *C. elegans* lysates, according to the manufacturer’s instructions. RNA quality was confirmed and quantified using agarose gel electrophoresis and a Nano Drop^®^ spectrophotometer (Thermo Fisher Scientific, Ottawa, ON, Canada), respectively. After DNase treatment, mRNA was reverse transcribed using a QuantiTect^®^ Reverse Transcription kit (Qiagen, Mississauga, ON, Canada) at 42 °C for 30 min. qRT-PCR was then conducted to record the gene expression for HSP 70A (GenBank Accession No. M18540); p53 upregulated modulator of apoptosis (PUMA, GenBank Accession No. AF354655); and BCL2 alpha (GenBank Accession No. NM_000633). This was accomplished using a Stratagene MX3005P LightCycler (La Jolla, CA, USA). The glyceraldehyde-3-phosphate dehydrogenase gene (GAPDH; GenBank Accession No. X04818) was used as the reference housekeeping gene for qRT-PCR analyses. The primer pairs used during qRT-PCR are listed in Table 2. qRT-PCR reactions were performed using QuantiFast^®^ SYBR^®^ Green PCR kit (Qiagen, Mississauga, ON, Canada) following the manufacturer’s instructions. The thermocycler conditions included an initial denaturation at 95 °C for 5 min, followed by 45 cycles of denaturation at 95 °C for 30 s, annealing at 55 °C for 30 s and elongation at 60 °C for 45 s, to amplify the target gene. Relative gene expression levels were corrected against the GAPDH housekeeping gene using MXPro software. All analyses were completed in triplicate.

### 2.6. Microarray Analysis of the Individual LOs

Microarray analysis was performed to determine LO effects on the expression of genes involved in regulating apoptosis (i.e., pro- or anti-apoptotic genes) in Calu-3 cells. These analyses were using an Rt2 Profiler PCR Super Array kit (Super Arrays Bioscience Corp., Frederick, MD, USA) following manufacturer’s instructions and were recorded using an MX3005P LightCycler (Stratagene, La Jolla, CA, USA).

### 2.7. Statistical Analysis

Statistical analyses were performed using the Statistical Package for the Social Sciences (SPSS) software *ver.* 18.0 (IBM Corp., Somers, NY, USA). One-way analysis of variance (ANOVA) and Duncan’s multiple-range tests were used to compare means within treatment groups. Data are presented as mean ± SD. Statistical significance was measured at 95% (*p* < 0.05) and 99% (*p* < 0.01).

## 3. Results and Discussion

### 3.1. Conformation of LOs in an Aqueous-DMSO Solution

The conformations of LOs were studied in a solution of aqueous-DMSO, as well as in the presence of the most used co-solvent for stabilizing conformation. CD spectroscopy identified secondary peptide structures present in LOs and showed that the LOs in solution were dominated by unordered structures, thus giving the ability to distinguish between the binding and inhibition of aggregation events. In an aqueous-DMSO solution, CD identified α-helices as the dominant secondary structures in LOs; although, there were more α-helices in this solvent than in other organic solvents (1,4-dioxane and tetrahydrofuran, data not shown). The CD spectra of LOB3 and LO1OB2 (with methionine sulfoxide) suggested high levels of α-helices and lesser amounts of other secondary structures compared to that of LO1OB1 (Table 3). However, overall, LO1OB2 and LO1OB1 with methionine residue still contained >75% of α-helices. The secondary structure of LOB3 was composed of α-helix (99.9%), parallel β-sheet (0.3%), β-turn (0.7%), and random coil conformations (0.1%). These observations were supported by a relatively good fit to the experimental spectra. LOs in aqueous-DMSO solutions indicated large amounts of hydrophobic compounds. However, none of these conformational forms appeared to be stabilized by strong intramolecular hydrogen bonds. The methods described in this study demonstrated greater recovery of hydrophobic cyclic peptides than other conventional methods.

### 3.2. Binding of LOs in Adipose Tissue

Diets containing either flaxseed or flaxseed oil were fed to pigs or rainbow trout, respectively, to investigate the distribution of LOs in these model organisms. After the consumption of extruded flaxseed or flaxseed oil, three LOs were identified by molecular ions in the mass spectra of animal fats (Figure 2). The trace amounts of hydrophobic LOs primarily distributed to the adipose tissues, as inferred by the presence of LO1OB1, LO1OB2, and LOB3. However, it is uncertain if enough LOs were absorbed to exert a biological response.

### 3.3. Induction of HSP 70A in Nematodes

Nematode cultures (*C. elegans*) were exposed to LOB3 at 0.1 μM, 1.0 μM, 10.0 μM and 100.0 μM for 2 h to monitor the effects of LOB3 on HSP 70A gene expression. At low concentrations (<1.0 μM), LOB3 induced a stress response in *C. elegans* cultures, signified by the production of HSP 70A (Figure 3). Interestingly, LOB3 illustrated an inverted U-shaped dose-response, in which the addition of either 0.1 µM or 10.0 µM of LOB3 to the culture media of *C. elegans* resulted in a 30% increase in HSP 70A protein production. With the addition of 1.0 µM LOB3, the production of this protein increased 3.5-fold (Figure 3). However, treatment at the highest concentration (e.g., 100 µM) was lethal. These results further support previous studies where no cytotoxic effects were observed at doses lower than 10 µM in cancer C6 cells and breast cancer cell line MDA-MB-231 [10]. At concentrations greater than 10.0 µM, cytotoxic effects were observed in C6 cells, including changes in cell shape and a reduction in cell numbers [10]. These findings suggest that structural deformities may have occurred in the highest dose group and thus resulted in mortality. Nonetheless, the administration of LOB3 at concentrations < 10.0 µM induced a strong stress response in *C. elegans*, inferred by the induction of HSP 70A. These proteins function in a variety of biochemical processes, including protein folding, transportation, and regulation of heat shock response [34]. The activation of the HSP 70 protein can also mediate cellular protection by interfering with the process of apoptotic cell death [35]. This is accomplished by blocking the assembly of a functional apoptosome and preventing apoptosis [35]. Furthermore, during the initial stages of tumorigenesis, HSP 70 can protect cells from oncogenic stress induced by the over-expression of oncogenes [36], as well as suppress cellular senescence, which is an important anti-tumor mechanism [37]. However, HSP 70 is also an essential factor in tumorigenesis [38], promoting tumor cell growth by inhibiting apoptosis and/or stabilizing the lysosomal membrane [39,40]. The expression of HSP 70 is also known to activate innate and adaptive immunity through the production of cytokines [41]. In addition, the increased translocation of HSP 70 into the extracellular milieu has also been observed to enhance the sensitivity of cancer cells to chemotherapies [41]. Due to these properties, it is important to ensure the induction of HSP 70A production by LOs is beneficial in exhibiting anti-cancer effects rather than contributing to tumorigenesis. Nonetheless, the administration of LOs to nematodes resulted in the activation of HSPs, suggesting its importance in elucidating a cellular response in mitigating stressors and maintaining homeostasis. Although, the identification of the regulatory pathways involved in controlling HSP 70A expression and activity should be further investigated [42]. This can lead to enhanced diagnoses or designs in developing new therapeutic strategies to combat cancerous cells, including the inhibition of multiple signaling pathways in cancer cells [41].

### 3.4. Induction of Apoptosis in Human Lung Cancer Cell Lines with LOs

The expressions of PUMA and the anti-apoptotic gene BCL2 both decreased with increasing concentrations (0.1–100 µM) of each of the three LOs (*p* < 0.05) (Figure 4). These results suggest that the gene expression of the pro-apoptotic gene PUMA was influenced by LOs. This supports earlier studies where cyclin-dependent kinase inhibitors and tumor-suppressor proteins were upregulated in the presence LOs [43], which can demonstrate anti-cancer properties, such as preventing the over-proliferation of cancer cells or mediating p53-induced cell death [44]. Interestingly, the expression of the anti-apoptotic gene BCL2 significantly declined with increased LO concentration than in CPT as a positive control (*p* < 0.05) (Figure 4B). Similar results have been observed in adenocarcinoma cells where the expression of Bax was upregulated while BCL2 expression was down-regulated in a dose-dependent manner [45]. Altogether, the up-regulation of apoptotic genes, and down-regulation of BCL2, induced by LOs, indicate potential applications of these compounds as chemotherapy agents due to their anti-cancer properties.

### 3.5. Microarray Analysis of the Individual LOs

Gene expression in human lung adenocarcinoma cells exposed to LOs was investigated by microarray analyses to determine effects on gene regulation and induction of apoptosis (Table 4). The results further supported our findings of the upregulation of apoptotic genes combined with the suppression of anti-apoptotic genes (i.e., BCL2). Comparatively, gene expression profiling of human lung epithelial cells (e.g., Calu-3) treated with RGDSK/K-RNT (5:50 mM) for 4 h induced over-expression of pro-apoptotic genes. Increased expression of BAK1, BCL2, CASP10, CIDEB, TP53BP2, Fas, TNF, and FasL suggest that p38 MAPK regulates RGDSK/K-RNT-induced apoptosis in Calu-3 cells. Furthermore, flaxseed administration has been demonstrated to reduce phosphorylated levels of MAPKs and suppress the expression of cytokines [24]. This suggests that LOs may be involved in the MAPK/AP-1 kinase signaling pathway and can be used in mitigating stressors. Although only LO1OB1, LO1OB2, and LOB3 were investigated in this study, it would be of interest to investigate the bioactive properties of other LOs. Nonetheless, the physicochemical and nutraceutical properties of LOs demonstrate the novelty of employing LOs for a variety of potential applications, including pharmaceutical, medicinal, and food supplement uses.

## 4. Conclusions

LOs are bioactive compounds isolated from flaxseed oil that can have significant potential for therapeutic applications. These compounds can be readily obtained as an added-value natural flaxseed product and have demonstrated significant immunosuppressing activities, including anti-cancer effects. This study further identified broad-spectrum bioactivities associated with LOs, including induction of HSP 70A production in *C. elegans* and apoptosis in human lung epithelial cancer lines. LOs were additionally involved in modulating regulatory genes associated with apoptosis in human lung epithelial cancer lines, suggesting their anti-cancer properties. Therefore, the regulation of apoptosis genes by LOs indicates the possibility of applying these compounds as a form of chemotherapy agent.

## Figures and Tables

**Figure 1 foods-11-03761-f001:**
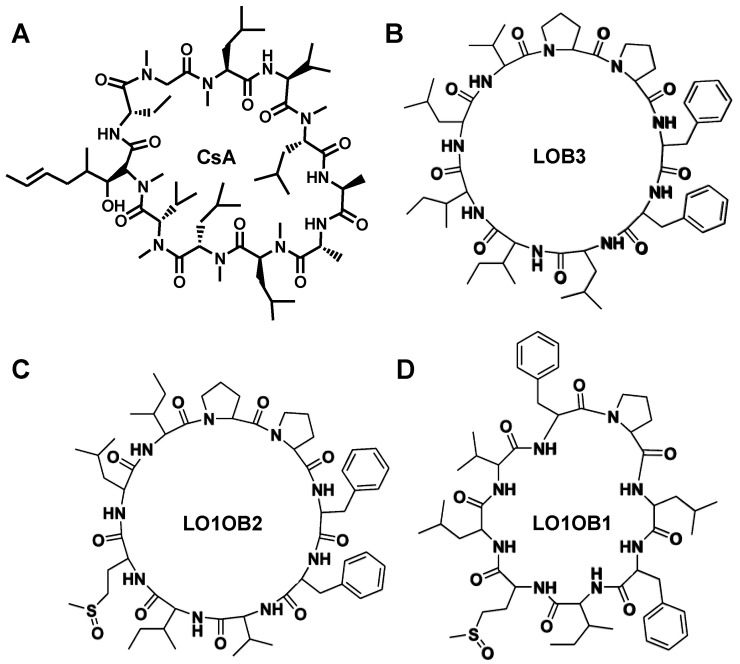
Chemical structures of (**A**) CsA and (**B**–**D**) LOs.

**Figure 2 foods-11-03761-f002:**
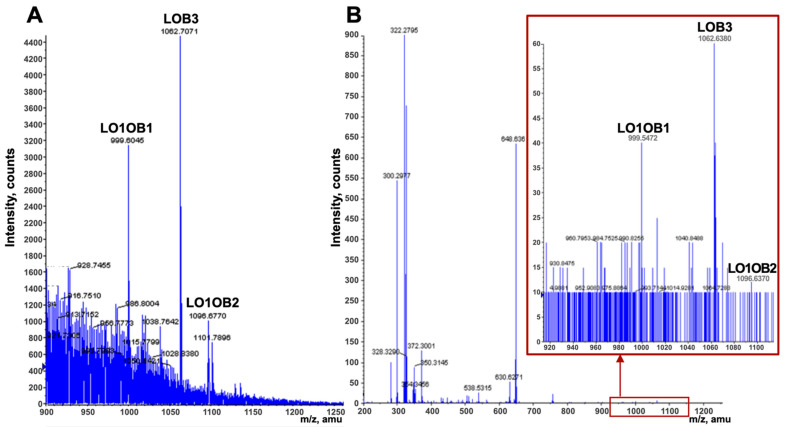
Localization of observed LOs (LO1OB1, LO1OB2, and LOB3) in (**A**) pig and (**B**) rainbow trout adipose, identified via direct injection on an Agilent 1100-series HPLC-ToF-MS.

**Figure 3 foods-11-03761-f003:**
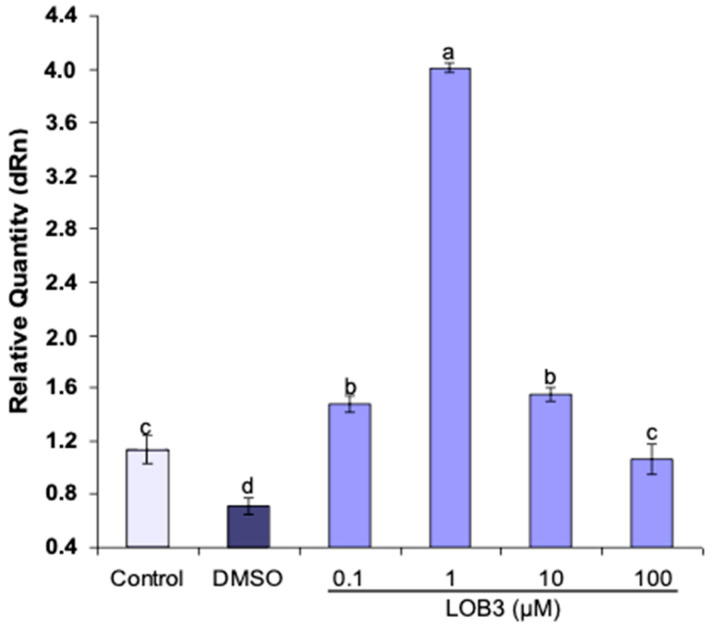
Induction of HSP 70A mRNA expression in *C. elegans* by LOB3 at different concentrations. The different letters (a–d) represent significance at *p* < 0.01. All samples were normalized to GAPDH expression. An Unamended culture medium and DMSO were used as the control treatment and negative control, respectively. Mean HSP 70 values and SDs from three repetitions shown. Data were analyzed by one-way ANOVA followed by Duncan’s multiple range test. Values not sharing a common letter are significantly different (*p* < 0.01).

**Figure 4 foods-11-03761-f004:**
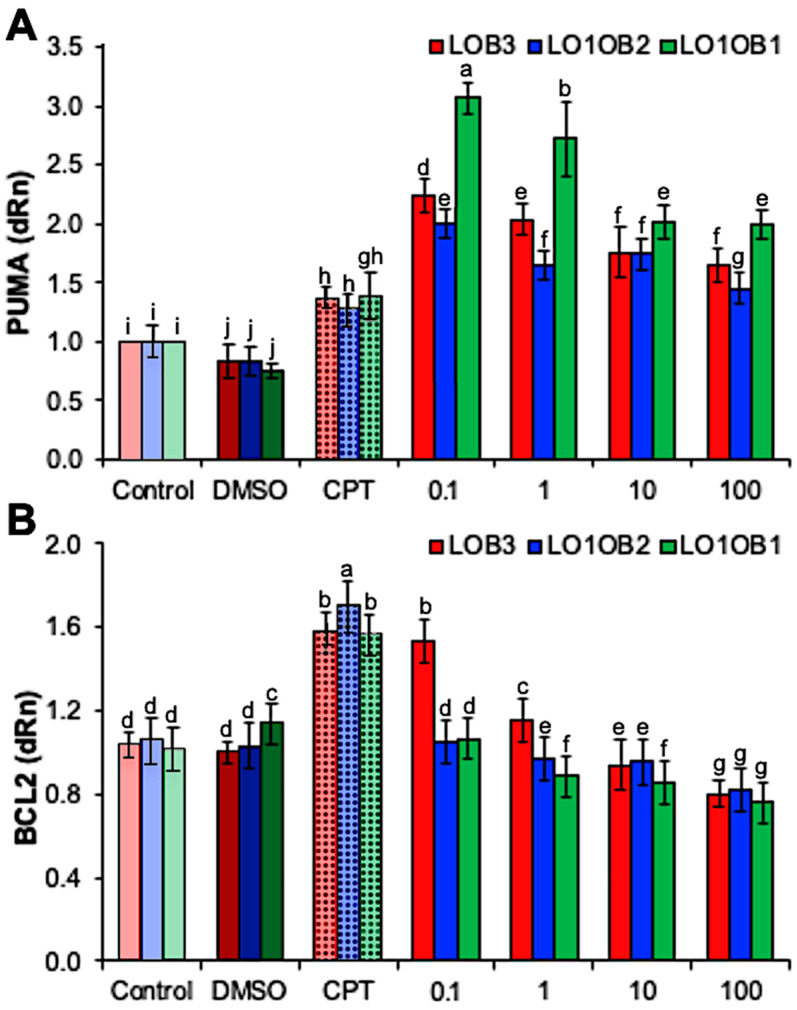
Gene expressions of (**A**) PUMA and (**B**) BCL2 after LO treatments. DMSO is a common negative control and is used as a solvent for dissolving LOs as LOs do not dissolve in water. CPT (positive control) is a cytotoxic quinoline alkaloid that inhibits the DNA enzyme topoisomerase I and has shown remarkable anti-cancer activity. The different letters (a–j) represent significance at *p <* 0.05.

**Table 1 foods-11-03761-t001:** Primary structures of select LOs.

Code ^a^	New Name ^b^	Literature Name ^c^	Molecular Weight (Da)	Chemical Formula
LOB3	[1–9-NαC]-linusorb B3	CLA	1040.34	C_57_H_85_N_9_O_9_
LO1OB2	[1–9-NαC], [1-(*R*_s_, *S*_s_)-MetO]- linusorb B2	CLC	1074.37	C_56_H_83_N_9_O_10_S
LO1OB1	[1–8-NαC], [1-(*R*_s_, *S*_s_)-MetO]- linusorb B1	CLE	977.26	C_51_H_76_N_8_O_9_S

^a^ Codes are O for methionine *S*-oxide; ^b^ The systematic nomenclature proposed by Shim et al. (2015) [29]; ^c^ Name used in subsequent literature description.

**Table 2 foods-11-03761-t002:** Primer pairs for investigated genes: PUMA, BCL2 alpha, and GAPDH.

cDNA	Forward Primer	Reverse Primer
PUMA	5′-ATG AAA TTT GGC ATG GGG TCT-3′	5′-GCC TGG TGG ACC GCC C-3′
BCL2 alpha	5′-ATG GCG CAC GCT GGG AGA AC-3′	5′-GCG ACC GGG TCC CGG GAT GC-3′
GAPDH	5′-ATG GGG AAG GTG AAG-3′	5′-GAC AAG CTT CCC GTT CTC-3′

**Table 3 foods-11-03761-t003:** Protein secondary structure estimates for LOs of CD spectra (260–200 nm) in an aqueous-DMSO solution.

Code	α-Helix (%)	Antiparallel (%)	Parallel (%)	β-Turn (%)	Random Coil (%)
LOB3	98.90	0.00	0.30	0.70	0.10
LO1OB2	98.90	0.00	0.40	0.60	0.10
LO1OB1	75.30	0.00	0.30	6.20	18.20

**Table 4 foods-11-03761-t004:** Curated list of genes up- or down-regulated in Calu-3 exposed to LOs.

Symbol	Description	LOB3	LO1OB2	LO1OB1	Role in Carcinogenesis
BAK1	BCL2 antagonist/killer 1	+ ^a^	NE ^b^	NE	Brassinosteroid signaling, light responses, cell death, and plant innate immunity [46]
BCL2	B-cell lymphoma 2	+	–	– ^c^	Caspase activation, regulation of apoptosis [47]
CASP10	Caspase 10	NE	+	NE	Aspartate-specific cysteine protease that participates in the apoptotic pathway [48]
CIDEA	Cell death-inducing DFFA- like effector A	NE	+	+	Apoptotic gene [49]
CIDEB	Cell death-inducing DFFA- like effector B	+	+	+	Apoptotic gene [49]
HRK	Harakiri	NE	+	+	Activates and regulates apoptosis, pro-apoptotic gene [50]
Fas	TNF receptor superfamily member 6	+	+	+	Induces apoptosis [51]
FasL	Fas ligand	+	NE	NE	Regulation of cell death [52]
TNF	Tumor necrosis factor	+	+	+	Cytokine that contributes to both physiological and pathological processes [53]
TP53	Tumor protein p53	+	+	NE	Tumor suppressor protein involved in preventing cancer [54]
TP53BP2	Tumor protein p53 binding protein 2	+	+	+	Regulates the proliferation, apoptosis, autophagy, migration, of tumor cells [55]

^a^ +: genes over-expressed; ^b^ NE: not expressed; ^c^ –: genes under-expressed.

## Data Availability

The data of the current study are available from the corresponding author on reasonable request.

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
