# Peer review of "Uptake of Flaxseed Dietary Linusorbs Modulates Regulatory Genes Including Induction of Heat Shock Proteins and Apoptosis"

_foods, 2022, doi:10.3390/foods11233761_

Round 1

Reviewer 1 Report

In general, I find this work interesting. It brings new findings about the anti-cancer properties of linusrobs (LOs), predetermining them as potential chemotherapy agents.

However, I have some suggestions for minor corrections which could improve the quality of the paper:

Title

The title of the manuscript sounds quite general. "Broad Spectrum Bioactivities… "implies that the manuscript is focused on the quite broad spectrum of bioactivities of LOs, like antioxidant, anti-inflammatory, anti-malarial or other general and known activities of LOs. However, the manuscript deals exclusively with the immunosuppressive activities or anti-cancer effects of these peptides, respectively. Thus I suggest renaming the title to become more pointed and corresponding to your research topic. 

Materials and Methods

In chapter 2.1, there is stated that you used "flaxseed meals". Does it mean using meals from more cultivars, or was it an unspecified mixture of flax meals? However, if it is known and possible, please specify the name of the cultivar as the more specific genetic source of the identified LOs for the eventual further comparisons with the other cultivars. Before the feeding experiment, it was the flaxseed meal somehow processed? If yes, it should be stated in the methods.

I strongly recommend providing the detailed methodology (in the main text or supplement) applied for the quantification of LOs by LC-MS (chapter 2.3), such as the model of column used for chromatographic separation and used separation temperature, parameters of gradient elution (if it was used) and specific, adjustable parameters used within MS analyses.

Tables

The first and the second tables in the manuscript are described as "Table 2". Please correct the numbering. In the first table of the manuscript, please consider replacing the parameter "formula weight" with "molecular weight" (more typical for peptides) and also state the corresponding unit type (Da).

Figures

Figures 3 and 4 are, unfortunately, blurry and aren't readable. It probably looks like some technical error, but it should definitely be corrected before eventual publication.

Results and discussion

Except for the mentioned figures, the results are clearly presented. I have just two points for the improvement of the manuscript. 

Concerning the activation of HSP70 by the LOB3, this effect must not be unambiguously positive. At the initial stages of tumorigenesis, HSP70 can protect cells undergoing transformation from oncogenic stress induced by overexpression of oncogenes. HSP70 has also been shown to suppress cellular senescence, which is an important anti-tumor mechanism at the early stages of tumorigenesis. On the other hand, HSP70 promote tumor cells growth and survival. The levels of HSP70 positively correlate with the development of cancer (see e.g. https://doi.org/10.1016/j.canlet.2012.06.003). Thus, the authors should clearly state why and when could be the induction of HSP 70A production by LOs beneficial from the point of anti-cancer effects.

The second recommendation is related to the role of the genes, stated in table 4, in carcinogenesis. Since the readers may not be familiar with these genes and their role, it would be helpful to state their clear relation(s) to cancer, preferably in table 4, with the corresponding literature references.

Conclusions

As in the title, I recommend avoiding the statement about investigating the broad-spectrum bioactivities related to this research. From my perspective, this manuscript focuses only on the quite narrowly related biological activities of LOs.

Author Response

Reviewer 1 Comments:

Comment: The title of the manuscript sounds quite general. "Broad Spectrum Bioactivities… "implies that the manuscript is focused on the quite broad spectrum of bioactivities of LOs, like antioxidant, anti-inflammatory, anti-malarial or other general and known activities of LOs. However, the manuscript deals exclusively with the immunosuppressive activities or anti-cancer effects of these peptides, respectively. Thus, I suggest renaming the title to become more pointed and corresponding to your research topic.

Response: The title has been changed to “Biological Activity of Flaxseed (Linum usitatissimum L.) Linusorbs” on line 2.

Comment: In chapter 2.1, there is stated that you used "flaxseed meals". Does it mean using meals from more cultivars, or was it an unspecified mixture of flax meals? However, if it is known and possible, please specify the name of the cultivar as the more specific genetic source of the identified LOs for the eventual further comparisons with the other cultivars. Before the feeding experiment, it was the flaxseed meal somehow processed? If yes, it should be stated in the methods. I strongly recommend providing the detailed methodology (in the main text or supplement) applied for the quantification of LOs by LC-MS (chapter 2.3), such as the model of column used for chromatographic separation and used separation temperature, parameters of gradient elution (if it was used) and specific, adjustable parameters used within MS analyses.

Response: The LOs were identified using HPLC equipped with a ToF-MS as described in Section 2.3. No column or gradient elution was used, as it was analyzed via direct infusion. The analysis of LOs using HPLC-ToF-MS was mainly used for qualification (e.g., identification/confirmation of LOs in the sample matrix).  We do agree, it would have been beneficial to quantify the presence of LOs in the adipose tissue, at the time; unfortunately, this was not done.

Comment:  The first and the second tables in the manuscript are described as "Table 2". Please correct the numbering. In the first table of the manuscript, please consider replacing the parameter "formula weight" with "molecular weight" (more typical for peptides) and also state the corresponding unit type (Da).

Response:  Thank you for catching that, these corrections have been applied on line 94.

Comment: Figures 3 and 4 are, unfortunately, blurry and aren't readable. It probably looks like some technical error, but it should definitely be corrected before eventual publication.

Response: Figures 3 and 4 are readable, there may have been an issue with the compression when it was uploaded, we will re-try submitting it.

Comment: Except for the mentioned figures, the results are clearly presented. I have just two points for the improvement of the manuscript. Concerning the activation of HSP70 by the LOB3, this effect must not be unambiguously positive. At the initial stages of tumorigenesis, HSP70 can protect cells undergoing transformation from oncogenic stress induced by overexpression of oncogenes. HSP70 has also been shown to suppress cellular senescence, which is an important anti-tumor mechanism at the early stages of tumorigenesis. On the other hand, HSP70 promote tumor cells growth and survival. The levels of HSP70 positively correlate with the development of cancer (see e.g., https://doi.org/10.1016/j.canlet.2012.06.003). Thus, the authors should clearly state why and when could be the induction of HSP70A production by LOs beneficial from the point of anti-cancer effects.

Response: Thank you for this suggestion! Additional information on HSP 70 activity during tumorigenesis and the interaction of LOs on HSP 70A has been further described in Section 3.3 on lines 241-259.

Comment: The second recommendation is related to the role of the genes, stated in table 4, in carcinogenesis. Since the readers may not be familiar with these genes and their role, it would be helpful to state their clear relation(s) to cancer, preferably in table 4, with the corresponding literature references.

Response: Thank you for the suggestion. We previously had the description written below Table 4., but now have included it in the text of Table 4 with accompanying references on lines 307-308.

Comment: As in the title, I recommend avoiding the statement about investigating the broad-spectrum bioactivities related to this research. From my perspective, this manuscript focuses only on the quite narrowly related biological activities of LOs.

Response: The title has been changed to “Biological Activity of Flaxseed (Linum usitatissimum L.) Linusorbs” on line 2.

Reviewer 2 Report

The manuscript deals with an important and fruitful topic. The results are suitable but more likely in a medical or biologically focused journal.

The authors need to improve the presentation of the results! The results presented in Figures 3 and 4 are not readable and must necessarily be improved (For this reason, I was unable to fully review the results). Figure 2 could have more description of the values (apparently this is output from the instrument operating software). 

Author Response

Reviewer 2 Comments:

Comment: The authors need to improve the presentation of the results! The results presented in Figures 3 and 4 are not readable and must necessarily be improved (For this reason, I was unable to fully review the results).

Response: Figures 3 and 4 are readable, there may have been an issue with the compression when it was uploaded, we will re-try submitting it.

Comment: Figure 2 could have more description of the values (apparently this is output from the instrument operating software).

Response: Figure 2 description has been expanded upon including detailing the presence of specific linusorbs in the adipose tissue of pig and rainbow trout.
